# Understanding the Representation Power of Graph Neural Networks in Learning Graph Topology

**Nima Dehmamy**[*]
CSSI, Kellogg School of Management
Northwestern University, Evanston, IL
nimadt@bu.edu

**Albert-László Barabási**[†]
Center for Complex Network Research,
Northeastern University, Boston MA
alb@neu.edu

**Rose Yu**
Khoury College of Computer Sciences,
Northeastern University, Boston, MA
roseyu@northeastern.edu

## Abstract

To deepen our understanding of graph neural networks, we investigate the representation power of Graph Convolutional Networks (GCN) through the looking glass of *graph moments*, a key property of graph topology encoding path of various lengths. We find that GCNs are rather restrictive in learning graph moments. Without careful design, GCNs can fail miserably even with multiple layers and nonlinear activation functions. We analyze theoretically the expressiveness of GCNs, concluding that a modular GCN design, using different propagation rules with residual connections could significantly improve the performance of GCN. We demonstrate that such modular designs are capable of distinguishing graphs from different graph generation models for surprisingly small graphs, a notoriously difficult problem in network science. Our investigation suggests that, depth is much more influential than width, with deeper GCNs being more capable of learning higher order graph moments. Additionally, combining GCN modules with different propagation rules is critical to the representation power of GCNs.

## 1 Introduction

The surprising effectiveness of graph neural networks [17] has led to an explosion of interests in graph representation learning, leading to applications from particle physics [12], to molecular biology [37] to robotics [4]. We refer readers to several recent surveys [7, 38, 33, 14] and the references therein for a non-exhaustive list of the research. Graph convolution networks (GCNs) are among the most popular graph neural network models. In contrast to existing deep learning architectures, GCNs are known to contain fewer number of parameters, can handle irregular grids with non-Euclidean geometry, and introduce relational inductive bias into data-driven systems. It is therefore commonly believed that graph neural networks can learn arbitrary representations of graph data.

Despite their practical success, most GCNs are deployed as black boxes feature extractors for graph data. It is not yet clear to what extent can these models capture different graph features. One prominent feature of graph data is *node permutation invariance*: many graph structures stay the same

---

[*]work done when at Center for Complex Network Research, Northeastern University, Boston, MA

[†]Center for Cancer Systems Biology, Dana Farber Cancer Institute, Boston MA, Brigham and Women's Hospital, Harvard Medical School, Boston MA, Center for Network Science, Central European University, Budapest, Hungary

under relabelling or permutations of the nodes. For instance, people in a friendship network may be following a similar pattern for making friends, in similar cultures. To satisfy permutation invariance, GCNs assign global parameters to all the nodes by which significantly simplifies learning. But such efficiency comes at the cost of expressiveness: GCNs are *not* universal function approximators [34]. We use GCN in a broader sense than in [20], allowing different propagation rules (see below (4)).

To obtain deeper understanding of graph neural networks, a few recent work have investigated the behavior of GCNs including expressiveness and generalizations. For example, [28] showed that message passing GCNs can approximate measurable functions in probability. [34, 24, 25] defined expressiveness as the capability of learning multi-set functions and proved that GCNs are at most as powerful as the Weisfeiler-Lehman test for graph isomorphism, but assuming GCNs with infinite number of hidden units and layers. [32] analyzed the generalization and stability of GCNs, which suggests that the generalization gap of GCNs depends on the eigenvalues of the graph filters. However, their analysis is limited to a single layer GCN for semi-supervised learning tasks. Up until now, the representation power of multi-layer GCNs for learning graph topology remains elusive.

In this work, we analyze the representation power of GCNs in learning graph topology using *graph moments*, capturing key features of the underlying random process from which a graph is produced. We argue that enforcing node permutation invariance is restricting the representation power of GCNs. We discover pathological cases for learning graph moments with GCNs. We derive the representation power in terms of number of hidden units (width), number of layers (depths), and propagation rules. We show how a modular design for GCNs with different propagation rules significantly improves the representation power of GCN-based architectures. We apply our modular GCNs to distinguish different graph topology from small graphs. Our experiments show that depth is much more influential than width in learning graph moments and combining different GCN modules can greatly improve the representation power of GCNs. [3]

In summary, our contributions in this work include:

- We reveal the limitations of graph convolutional networks in learning graph topology. For learning graph moments, certain designs GCN completely fails, even with multiple layers and non-linear activation functions.

- we provide theoretical guarantees for the representation power of GCN for learning graph moments, which suggests a strict dependence on the depth whereas the width plays a weaker role in many cases.

- We take a modular approach in designing GCNs that can learn a large class of node permutation invariant function of of the graph, including non-smooth functions. We find that having different graph propagation rules with residual connections can dramatically increase the representation power of GCNs.

- We apply our approach to build a "graph stethoscope": given a graph, classify its generating process or topology. We provide experimental evidence to validate our theoretical analysis and the benefits of a modular approach.

**Notation and Definitions**   A graph is a set of $N$ nodes connected via a set of edges. The adjacency matrix of a graph $A$ encodes graph topology, where each element $A_{ij}$ represents an edge from node $i$ to node $j$. We use $AB$ and $A \cdot B$ (if more than two indices may be present) to denote the matrix product of matrices $A$ and $B$. All multiplications and exponentiations are matrix products, unless explicitly stated. Lower indices $A_{ij}$ denote $i, j$th elements of $A$, and $A_i$ means the $i$th row. $A^p$ denotes the $p$th matrix power of $A$. We use $a^{(m)}$ to denote a parameter of the $m$th layer.

## 2   Learning Graph Moments

Given a collection of graphs, produced by an unknown random graph generation process, learning from graphs requires us to accurately infer the characteristics of the underlying generation process. Similar to how moments $\mathbb{E}[X^p]$ of a random variable $X$ characterize its probability distribution, graph moments [5, 23] characterize the random process from which the graph is generated.

## 2.1 Graph moments

In general, a $p$th order graph moment $M_p$ is the ensemble average of an order $p$ polynomial of $A$

$$M_p(A) = \prod_{q=1}^{p} (A \cdot W_q + B_q) \tag{1}$$

with $W_q$ and $B_q$ being $N \times N$ matrices. Under the constraint of node permutation invariance, $W_q$ must be either proportional to the identity matrix, or a uniform aggregation matrix. Formally,

$$M(A) = A \cdot W + B, \quad \text{Node Permutation Invariance} \Rightarrow \quad W, B = cI, \quad \text{or} \quad W, B = c\mathbf{1}\mathbf{1}^T \tag{2}$$

where $\mathbf{1}$ is a vector of ones. Graph moments encode topological information of a graph and are useful for graph coloring and Hamiltonicity. For instance, graph power $A_{ij}^p$ counts the number of paths from node $i$ to $j$ of length $p$. For a graph of size $N$, $A$ has $N$ eigenvalues. Applying eigenvalue decomposition to graph moments, we have $\mathbb{E}[A^p] = \mathbb{E}[(V^T \Lambda U)^p]) = V^T \mathbb{E}[\Lambda^p] U$. Graphs moments correspond to the distribution of the eigenvalues $\Lambda$, which are random variables that characterize the graph generation process. Graph moments are node permutation invariant, meaning that relabelling of the nodes will not change the distribution of degrees, the paths of a given length, or the number of triangles, to name a few. The problem of learning graph moments is to learn a functional approximator $F$ such that $F : A \rightarrow M_p(A)$, while preserving node permutation invariance.

Different graph generation processes can depend on different orders of graph moments. For example, in Barabási-Albert (BA) model [1], the probability of adding a new edge is proportional to the degree, which is a first order graph moment. In diffusion processes, however, the stationary distribution depends on the normalized adjacency matrix $\hat{A}$ as well as its symmetrized version $\hat{A}_s$, defined as follows:

$$D_{ij} \equiv \delta_{ij} \sum_k A_{ik} \qquad \hat{A} \equiv D^{-1}A \qquad \hat{A}_s \equiv D^{-1/2}AD^{-1/2} \tag{3}$$

which are *not* smooth functions of $A$ and have no Taylor expansion in $A$, because of the inverse $D^{-1}$. Processes involving $D^{-1}$ and $A$ are common and per (2) $D$ and $\text{Tr}[A]$ are the only node permutation invariant first order moments of $A$. Thus, in order to approximate more general node permutation invariant $F(A)$, it is crucial for a graph neural network to be able to learn moments of $A$, $\hat{A}$ and $\hat{A}_s$ simultaneously. In general, non-smooth functions of $A$ can depend on $A^{-1}$, which may be important for inverting a diffusion process. We will only focus on using $A$, $\hat{A}$ and $\hat{A}_s$ here, but all argument hold also if we include $A^{-1}$, $\hat{A}^{-1}$ and $\hat{A}_s^{-1}$ as well.

## 2.2 Learning with Fully Connected Networks

Consider a toy example of learning the first order moment. Given a collection of graphs with $N = 20$ nodes, the inputs are their adjacency matrices $A$, and the outputs are the node degrees $D_i = \sum_{j=1}^{N} A_{ij}$. For a fully connected (FC) neural network, it is a rather simple task given its universal approximation power [19]. However, a FC network treats the adjacency matrices as vector inputs and ignores the underlying graph structures, it needs a large amount of training samples and many parameters to learn properly.

Fig. 1 shows the mean squared error (MSE) of a single layer FC network in learning the first order moments. Each curve corresponds to different number of training samples, ranging from 500–10,000. The horizontal axis shows the number of hidden units. We can see that even though the network can learn the moments properly reaching an MSE of $\approx 10^{-4}$, it requires the same order of magnitude of hidden units as the number of nodes in the graph, and at least $1,000$ samples. Therefore, FC networks are quite inefficient for learning graph moments, which motivates us to look into more power alternatives: graph convolution networks.

## 2.3 Learning with Graph Convolutional Networks

We consider the following class of graph convolutional networks. A single layer GCN propagates the node attributes $h$ using a function $f(A)$ of the adjacency matrix and has an output given by

$$F(A, h) = \sigma\left(f(A) \cdot h \cdot W + b\right) \tag{4}$$

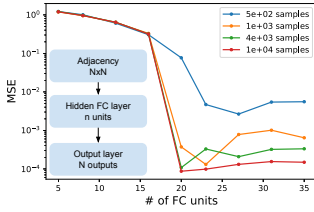

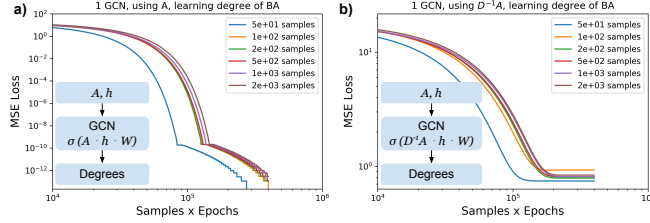

Figure 1: Learning graph moments (Erdős-Rényi graph) with a single fully-connected layer. Best validation MSE w.r.t number of hidden units $n$ and the number of samples in the training data (curves of different colors).

Figure 2: Learning the degree of nodes in a graph with a single layer of GCN. When the GCN layer is designed as $\sigma(A \cdot h \cdot W)$ with linear activation function $\sigma(x) = x$, the network easily learns the degree (a). However, if the network uses the propagation rule as $\sigma(D^{-1}A \cdot h \cdot W)$, it fails to learn degree, with very high MSE loss (b). The training data were instances of Barabasi-Albert graphs (preferential attachment) with $N = 20$ nodes and $m = 2$ initial edges.

where $f$ is called the propagation rule, $h_i$ is the attribute of node $i$, $W$ is the weight matrix and $b$ is the bias. As we are interested in the graph topology, we ignore the node attributes and set $h_i = 1$. Note that the weights $W$ are only coupled to the node attributes $h$ but not to the propagation rule $f(A)$. The definition in Eqn (4) covers a broad class of GCNs. For example, GCN in [20] uses $f = D^{-1/2}AD^{-1/2}$. GraphSAGE [16] mean aggregator is equivalent to $f = D^{-1}A$. These architectures are also special cases of Message-Passing Neural Networks [13].

We apply a single layer GCN with different propagation rules to learn the node degrees of BA graphs. With linear activation $\sigma(x) = x$, the solution for learning node degrees is $f(A) = A$, $W = 1$ and $b = 0$. For high-order graph moments of the form $M_p = \sum_j (A^p)_{ij}$, a single layer GCN has to learn the function $f(A) = A^p$. As shown in Figure 2, a single layer GCN with $f(A) = A$ can learn the degrees perfectly even with as few as 50 training samples for a graph of $N = 20$ nodes (Fig. 2a). Note that GCN only requires 1 hidden unit to learn, which is much more efficient than the FC networks. However, if we set the learning target as $f(A) = D^{-1}A$, the same GCN completely fails at learning the graph moments regardless of the sample size, as shown in Fig. 2b. This demonstrates the limitation of GCNs due to the permutation invariance constraint. Next we analyze this phenomena and provide theoretical guarantees for the representation power of GCNs.

## 3 Theoretical Analysis

To learn graph topology, fully connected layers require a large number of hidden units. The following theorem characterizes the representation power of fully connected neural network for learning graph moments in terms of number of nodes $N$, order of moments $p$ and number of hidden units $n$.

**Theorem 1.** *A fully connected neural network with one hidden layer requires $n > O(C_f^2) \sim O(p^2 N^{2q})$ number of neurons in the best case with $1 \leq q \leq 2$ to learn a graph moment of order $p$ for graphs with $N$ nodes. Additionally, it also needs $S > O(nd) \sim O\left(p^2 N^{2q+2}\right)$ number of samples to make the learning tractable.*

Clearly, if a FC network fully parameterizes every element in a $N \times N$ adjacency matrix $A$, the dimensions of the input would have to be $d = N^2$. If the FC network allows weight sharing among nodes, the input dimension would be $d = N$. The Fourier transform of a polynomial function of order $p$ with $O(1)$ coefficients will have an $L_1$ norms of $C_f \sim O(p)$. Using Barron's result [2] with $d = N^q$, where $1 \leq q \leq 2$ and set the $C_f \sim O(p)$, we can obtain the approximation bound.

In contrast to fully connected neural networks, graph convolutional networks are more efficient in learning graph moments. A graph convolution network layer without bias is of the form:

$$F(A, h) = \sigma(f(A) \cdot h \cdot W) \tag{5}$$

Permutation invariance restricts the weight matrix $W$ to be either proportional to the identity matrix, or a uniform aggregation matrix, see Eqn. (2). When $W = cI$, the resulting graph moment $M_p(A)$ has exactly the form of the output of a $p$ layer GCN with linear activation function.

We first show, via an explicit example, that a $n < p$ layer GCN by stacking layers of the form in Eqn. (5) cannot learn $p$th order graph moments.

**Lemma 1.** *A graph convolutional network with $n < p$ layers cannot, in general, learn a graph moment of order $p$ for a set of random graphs.*

We prove this by showing a counterexample. Consider a directed graph of two nodes with adjacency matrix $A = \begin{pmatrix} 0 & a \\ b & 0 \end{pmatrix}$. Suppose we want to use a single layer GCN to learn the second order moment $f(A)_i = \sum_j (A^2)_{ij} = \sum_k A_{ik} D_k$. The node attributes $h_{il}$ are decoupled from the propagation rule $f(A)_i$. Their values are set to ones $h_{il} = 1$, or any values independent of $A$. The network tries to learn the weight matrix $W_{l\mu}$ and has an output $h^{(1)}$ of the form

$$h_{i\mu}^{(1)} = \sigma \left( A \cdot h \cdot W \right)_{i\mu} = \sigma \left( \sum_{j,l} A_{ij} h_{jl} W_{l\mu} \right), \tag{6}$$

For brevity, define $V_{i\mu} \equiv \sum_l h_{il} W_{l\mu}$. Setting the output $h^{(1)}$ to the desired function $A \cdot D$, with components $h_{1\mu}^{(1)} = h_{2\mu}^{(1)} = ab$, (hence $\mu$ can only be 1) and plugging in $A$, the two components of the output become

$$h_{1\mu}^{(1)} = \sigma \left( D_1 V_{1\mu} \right) = \sigma \left( a V_{1\mu} \right) = ab \qquad h_{2\mu}^{(1)} = \sigma \left( D_2 V_{2\mu} \right) = \sigma \left( b V_{2\mu} \right) = ab. \tag{7}$$

which must be satisfied $\forall a, b$. But it's impossible to satisfy $\sigma \left( a V_{1\mu} \right) = ab$ for $(a, b) \in \mathbb{R}^2$ with $V_{1\mu}$ and $\sigma(\cdot)$ independent of $a, b$. $\qquad \square$

**Proposition 1.** *A graph convolutional network with $n$ layers, and no bias terms, in general, can learn $f(A)_i = \sum_j (A^p)_{ij}$ only if $n = p$ or $n > p$ if the bias is allowed.*

If we use a two layer GCN to learn a first order moment $f(A)_i = \sum_j A_{ij} = D_i$, for the output of the second layer $h_{i\nu}^{(2)}$ we have

$$h^{(2)} = \sigma^{(2)} \left( A \cdot \sigma^{(1)} \left( A \cdot h \cdot W^{(1)} \right) \cdot W^{(2)} \right), \; h_{1\nu}^{(2)} = \sigma^{(2)} \left( a \sum_\mu \sigma^{(1)} \left( b V_{2\mu}^{(1)} \right) W_{\mu\nu}^{(2)} \right) = a \tag{8}$$

Again, since this must hold for any value of $a, b$ and $\nu$, we see that $h_{1\nu}^{(2)}$ is a function of $b$ through the output of the first layer $h_{2\mu}^{(1)}$. Thus $h_{1\nu}^{(2)} = a$ can only be satisfied if the first layer output is a constant. In other words, only if the first layer can be bypassed (e.g. if the bias is large and weights are zero) can a two-layer GCN learn the first order moment. $\qquad \square$

This result also generalizes to multiple layers and higher order moments in a straightforward fashion. For GCN with linear activation, a similar argument shows that when the node attributes $h$ are not implicitly a function of $A$, in order to learn the function $\sum_j (A^p)_{ij}$, we need to have exactly $n = p$ GCN layers, without bias. With bias, a feed-forward GCN with $n > p$ layers can learn single term order $p$ moments such as $\sum_j (A^p)_{ij}$. However, since it needs to set the some weights of $n - p$ layers to zero, it can fail in learning mixed order moments such as $\sum_j (A^q + A^p)_{ij}$.

To allow GCNs with very few parameters to learn mixed order moments, we introduce residual connections [18] by concatenating the output of every layer $[h^{(1)}, \ldots, h^{(m)}]$ to the final output of the network. This way, by applying an aggregation layer or a FC layer which acts the same way on the output for every node, we can approximate any polynomial function of graph moments. Specifically, the final $N \times d^o$ output $h^{(final)}$ of the aggregation layer has the form

$$h_{i\mu}^{(final)} = \sigma \left( \sum_{m=1}^{n} a_\mu^{(m)} \cdot h_i^{(m)} \right), \qquad h^{(m)} = \sigma(A \cdot h^{(m-1)} \cdot W^{(m)} + b^{(m)}), \tag{9}$$

where $\cdot$ acts on the output channels of each output layers. The above results lead to the following theorem which guarantees the representation power of multi-layer GCNs with respect to learning graph moments.

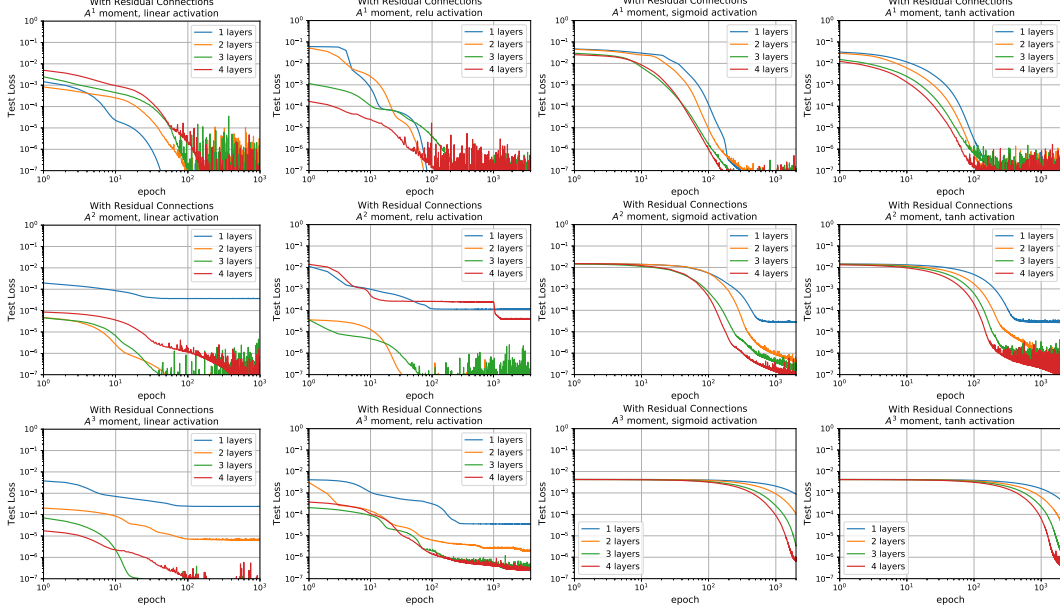

Figure 4: Test loss over number of epochs for learning first (top), second (middle) and third (bottom) order graph moments $M_p(A) = \sum_j (A^p)_{ij}$, with varying number of layers and different activation functions. A multi-layer GCN with residual connections is capable of learning the graph moments when the number of layers is at least the target order of the graph moments. The graphs are from our synthetic graph dataset described in Sec. 6.

**Theorem 2.** *With the number of layers $n$ greater or equal to the order $p$ of a graph moment $M_p(A)$, graph convolutional networks with residual connections can learn a graph moment $M_p$ with $O(p)$ number of neurons, independent of the size of the graph.*

Theorem 2 suggests that the representation power of GCN has a strong dependence on the number of layers (depth) rather than the size of the graph (width). It also highlights the importance of residual connections. By introducing residual connections into multiple GCN layers, we can learn any polynomial function of graph moments with linear activation. Interestingly, Graph Isomophism Network (GIN) proposed in [34] uses the following propagation rule:

$$F(A, h) = \sigma\left([(1 + \epsilon)I + A] \cdot h \cdot W\right) \tag{10}$$

which is a special case of our GCN with one residual connection between two modules.

## 4 Modular GCN Design

In order to overcome the limitation of the GCNs in learning graph moments, we take a modular approach to GCN design. We treat different GCN propagation rules as different "modules" and consider three important GCN modules (1) $f_1 = A$ [22] (2) $f_2 = D^{-1}A$ [20], and (3) $f_3 = D^{-1/2}AD^{-1/2}$ [16]. Figure 3a) shows the design of a single GCN layer where we combine three different GCN modules. The output of the modules are concatenated and fed into a node-wise FC layer. Note that our design is different from the multi-head attention mechanism in Graph Attention Network [31] which uses the same propagation rule for all the modules.

However, simply stacking GCN layers on top of each other in a feed-forward fashion is quite restrictive, as shown by our theoretical analysis for multi-layer GCNs. Different

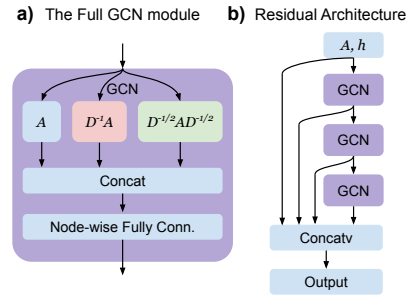

Figure 3: GCN layer (a), using three different propagation rules and a node-wise FC layer. Using residual connections (b) allows a $n$-layer modular GCN to learn any polynomial function of order $n$ of its constituent operators.

propagation rules cannot be written as Taylor expansions of each other, while all of them are important in modeling the graph generation process. Hence, no matter how many layers or how non-linear the activation function gets, multi-layer GCN stacked in a feed-forward way cannot learn network moments whose order is not precisely the number of layers. If we add residual connections from the output of every layer to the final aggregation layer, we would be able to approximate any polynomial functions of graph moments. Figure 3b) shows the design of a muli-layer GCN with residual connections. We stack the modular GCN layer on top of each other and concatenate the residual connections from every layer. The final layer aggregates the output from all previous layers, including residual connections.

We measure the representation power of GCN design in learning different orders of graph moments $M_p(A) = \sum_j (A^p)_{ij}$ with $p = 1, 2, 3$. Figure 4 shows the test loss over number of epochs for learning first (top), second (middle) and third (bottom) order graph moments. We vary the number of layers from 1 to 4 and test with different activation functions including linear, ReLU, sigmoid and tanh. Consistent with the theoretical analysis, we observe that whenever the number of layers is at least the target order of the graph moments, a multi-layer GCN with residual connections is capable of learning the graph moments. Interestingly, Jumping Knowledge (JK) Networks [35] showed similar effects of adding residual connections for Message Passing Graph Neural Networks.

Our modular approach demonstrates the importance of architectural design when using specialized neural networks. Due to permutation invariance, feed-forward GCNs are quite limited in their representation power and can fail at learning graph topology. However, with careful design including different propagation rules and residual connections, it is possible to improve the representation power of GCNs in order to capture higher order graph moments while preserving permutation invariance.

## 5 Related Work

**Graph Representation Learning**    There has been increasing interest in deep learning on graphs, see e.g. many recent surveys of the field [7, 38, 33]. Graph neural networks [22, 20, 17] can learn complex representations of graph data. For example, Hopfield networks [28, 22] propagate the hidden states to a fixed point and use the steady state representation as the embedding for a graph; Graph convolution networks [8, 20] generalize the convolutional operation from convolutional neural networks to learn from geometric objects beyond regular grids. [21] proposes a deep architecture for long-term forecasting of spatiotemporal graphs. [37] learns the representations for generating random graphs sequentially using an adversarial loss at each step. Despite practical success, deep understanding and theoretical analysis of graph neural networks is still largely lacking.

**Expressiveness of Neural Networks**    Early results on the expressiveness of neural networks take a highly theoretical approach, from using functional analysis to show universal approximation results [19], to studying network VC dimension [3]. While these results provided theoretically general conclusions, they mostly focus on single layer shallow networks. For deep fully connected networks, several recent papers have focused on understanding the benefits of depth for neural networks [11, 29, 28, 27]) with specific choice of weights. For graph neural networks, [34, 24, 25] prove the equivalence of a graph neural network with Weisfeiler-Lehman graph isomorphism test with infinite number of hidden layers. [32] analyzes the generalization and stability of GCNs, which depends on eigenvalues of the graph filters. However, their analysis is limited to a single layer GCN in the semi-supervised learning setting. Most recently, [10] demonstrates the equivalence between infinitely wide multi-layer GNNs and Graph Neural Tangent Kernels, which enjoy polynomial sample complexity guarantees.

**Distinguishing Graph Generation Models**    Understanding random graph generation processes has been a long lasting interest of network analysis. Characterizing the similarities and differences of generation models has applications in, for example, graph classification: categorizing a collections of graphs based on either node attributes or graph topology. Traditional graph classification approaches rely heavily on feature engineering and hand designed similarity measures [30, 15]. Several recent work propose to leverage deep architecture [6, 36, 9] and learn graph similarities at the representation level. In this work, instead of proposing yet another deep architecture for graph classification, we provide insights for the representation power of GCNs using well-known generation models. Our insights can provide guidance for choosing similarity measures in graph classification.

# 6 Graph Stethoscope: Distinguishing Graph Generation Models

An important application of learning graph moments is to distinguish different random graph generation models. For random graph generation processes like the BA model, the asymptotic behavior ($N \to \infty$) is known, such as scale-free. However, when the number of nodes is small, it is generally difficult to distinguish collections of graphs with different graph topology if the generation process is random. Thus, building an efficient tool that can probe the structure of small graphs of $N < 50$ like a stethoscope can be highly challenging, especially when all the graphs have the same number of nodes and edges.

**BA vs. ER.** We consider two tasks for graph stethoscope. In the first setting, we generate $5,000$ graphs with the same number of nodes and vary the number of edges, half of which are from the Barabasi-Albert (BA) model and the other half from the Erdos-Renyi (ER) model. In the BA model, a new node attaches to $m$ existing nodes with a likelihood proportional to the degree of the existing nodes. The $2,500$ BA graphs are evenly split with $m = 1, N/8, N/4, 3N/8, N/2$. To avoid the bias from the order of appearance of nodes caused by preferential attachment, we shuffle the node labels. ER graphs are random undirected graphs with a probability $p$ for generating every edge. We choose four values for $p$ uniformly between $1/N$ and $N/2$. All graphs have similar number of edges.

**BA vs. Configuration Model** One might argue that distinguishing BA from ER for small graphs is easy as BA graphs are known to have a power-law distribution for the node degrees [1], and ER graphs have a Poisson degree distribution. Hence, we create a much harder task where we compare BA graphs with "fake" BA graphs where the nodes have the same degree but all edges are rewired using the Configuration Model [26] (Config.). The resulting graphs share exactly the same degree distribution. We also find that higher graph moments of the Config BA are difficult to distinguish from real BA, despite the Config. model not fixing these moments.

Distinguishing BA and Config BA is very difficult using standard methods such as a Kolmogorov-Smirnov (KS) test. KS test measures the distributional differences of a statistical measure between two graphs and uses hypothesis testing to identify the graph generation model. Figure 5 shows the KS test values for pairs of real-real BA (blue) and pairs of real-fake BA (orange) w.r.t different graph moments. The dashed black lines show the mean of the KS test values for real-real pairs. We observe that the distributions of differences in real-real pairs are almost the same as those of real-fake pairs, meaning the variability in different graph moments among real BA graphs is almost the same as that between real and Config BA graphs.

Table 1: Test accuracy with different modules combinations for BA-ER. $f_1 = A$, $f_2 = D^{-1}A$, and $f_3 = D^{-1/2}AD^{-1/2}$.

| Modules | Accuracy |
|:---:|:---:|
| $f_1$ | 53.5 % |
| $f_3$ | 76.9 % |
| $f_1, f_3$ | 89.4 % |
| $f_1, f_2, f_3$ | 98.8 % |

**Classification Using our GCN Module** We evaluate the classification accuracy for these two settings using the modular GCN design, and analyze the trends of representation power w.r.t network depth and width, as well as the number of nodes in the graph. Our architecture consists of layers of our GCN module (Fig. 3, linear activation). The output is passed to a fully connected layer with softmax activation, yielding and $N \times c$ matrix ($N$ nodes in graph, $c$ label classes). The final

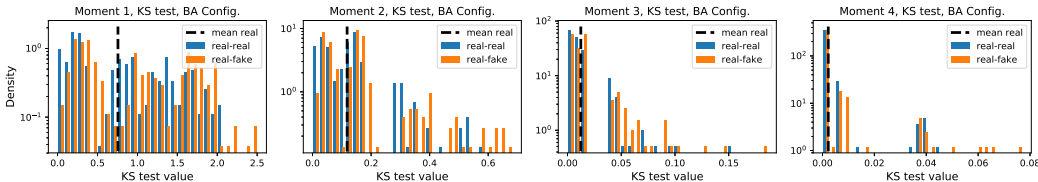

Figure 5: Distribution of Kolmogorov-Smirnov (KS) test values for differences between graph the first four graph moments $\sum_i (A^p)_{ij}$ in the dataset. "real-real" shows the distribution of KS test when comparing the graph moments of two real instances of the BA. All graphs have $N = 30$ nodes, but varying number of links. The "real-fake" case does the KS test for one real BA against one fake BA created using the configuration model.

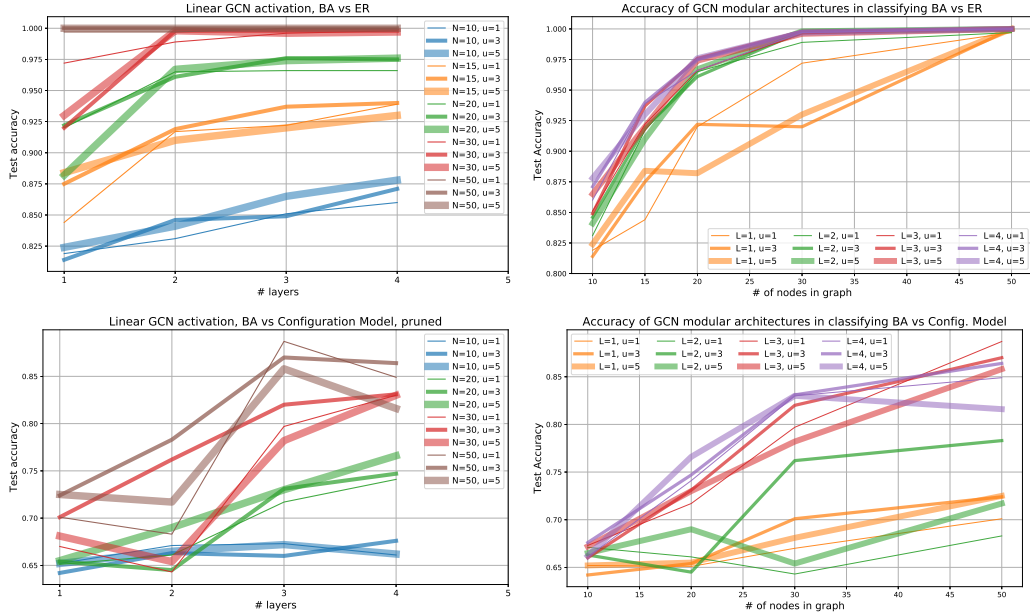

Figure 6: Classify graphs of **Barabasi-Albert** model vs. **Erdos-Renyi** model (top) and **Barabasi-Albert** model vs. **configuration** model (bottom). Left: test accuracy with respect to network depth for different number of nodes (N) and number of units (U). Right: test accuracy with respect to graph size for different number of layers (L) and number of units (U).

classification is found by mean-pooling over the $N$ outputs. We used mean-pooling to aggregate node-level representations, after which a single number is passed to a classification layer. Figure 6 left column shows the accuracy with increasing number of layers for different number of layers and hidden units. We find that depth is more influential than width: increasing one layer can improve the test accuracy by at least $5\%$, whereas increasing the width has very little effect. The right column is an alternative view with increasing size of the graphs. It is clear that smaller networks are harder to learn, while for $N \geq 50$ nodes is enough for $100\%$ accuracy in BA-ER case. BA-Config is a much harder task, with the highest accuracy of $90\%$.

We also conduct ablation study for our modular GCN design. Table 1 shows the change of test accuracy when we use different combinations of modules. Note that the number of parameters are kept the same for all different design. We can see that a single module is not enough to distinguish graph generation models with an accuracy close to random guessing. Having all three modules with different propagation rules leads to almost perfect discrimination between BA and ER graphs. This demonstrates the benefits of combining GCN modules to improve its representation power.

## 7   Conclusion

We conduct a thorough investigation in understanding what can/cannot be learned by GCNs. We focus on graph moments, a key characteristic of graph topology. We found that GCNs are rather restrictive in learning graph moments, and multi-layer GCNs cannot learn graph moments even with nonlinear activation. Theoretical analysis suggests a modular approach in designing graph neural networks while preserving permutation invariance. Modular GCNs are capable of distinguishing different graph generative models for surprisingly small graphs. Our investigation suggests that, for learning graph moments, depth is much more influential than width. Deeper GCNs are more capable of learning higher order graph moments. Our experiments also highlight the importance of combining GCN modules with residual connections in improving the representation power of GCNs.

## Acknowledgments

This work was supported in part by NSF #185034, ONR-OTA (N00014-18-9-0001).

## Footnotes

[3]   All   code   and   hyperparameters   are   available   at   `https://github.com/nimadehmamy/Understanding-GCN`

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
