[Supplementary Material]

# A Learning on graphs using single hidden layer fully connected network

**Proof of Theorem 1.** A fully connected neural network with sigmoid activation function $f_n$, in principle, could approximate any function $f$, provided there is enough training data. Barron's result [2] states that the upper-bound on the approximation error of a single layer is given by

$$\epsilon := \|f - f_n\| \sim O\left(\frac{C_f^2}{n}\right) + O\left(\frac{nd}{S}\log S\right), \tag{11}$$

where $n$ is the number of neurons, $d$ is the dimension of the input, $S$ is the number of samples, and $C_f$ is the $L_1$ norm of the Fourier coefficients of the function $f$.

$$C_f = \int d^d w |w|_1 |\tilde{f}(w)| \tag{12}$$

$$f(x) = \int d^d w e^{iwx} \tilde{f}(w)$$

$$|w|_1 = \sum_{j=1}^{d} |w_j|.$$

Using (11), we can bound the approximation error of for learning graph moments. Assume that the input is a graph with $N$ nodes, represented by an adjacency matrix $A$, the dimension of the input is thus $d = N^2$. If the number of nodes is not too large ($\log N \sim O(1)$), the second term in (11) essentially states that we need $S \sim O(N^2)$ samples to approximate any function well and avoid overfitting. The first term in (11) depends on the form of the function $f$. Specifically, $C_f$ depends on the Fourier coefficients which have a non-negligible magnitude. Consider for example a polynomial function $f(x) = \sum_{k=0}^{p} c_k x^k$ of order $p$ of a single variable $x$ defined over the unit interval $I = [-1, 1]$, so that it is Lebesgue integrable. Performing the Fourier transform over this interval yields a Fourier series with coefficients given by

$$f(x) = \sum_{k=0}^{p} c_k x^k = \sum_{m=0} \tilde{f}(m) e^{-2\pi i m x}$$

$$\tilde{f}(m) = \sum_{k=0}^{p} c_k \int_{-1}^{1} \frac{dx}{2\pi i} x^j e^{-2\pi i m x} = \frac{c_k}{2\pi i k!} \frac{\partial^k}{\partial m^k} \delta(m). \tag{13}$$

If all the coefficients $c_k \sim O(1)$, (13) states that at most $p$ Fourier coefficients will have $O(1)$ magnitudes for this polynomial function and so $C_f \sim O(p)$ for a polynomial $f$ of a single variable $x$. If $x$ is $d$ dimensional, we have $C_f \sim O(pd)$. We want to learn graph moments, which are polynomial functions of the elements in $A_{ij}$. For example, the node degree is a first order polynomial of the form $f(A)_i = \sum_{j=0}^{N} A_{ij}$. Higher order moments are generally functions of higher powers of $A$. For example, the number paths of length two between nodes nodes $i$ and $j$ on an unweighted graph are given by $P_{ij} = \sum_k A_{ik} A_{kj}$. We can write this as a second order function in $A_{ij}$

$$P_{ij} = \sum_{k,l=1}^{N} A_{ik} A_{lj} c^{kl}$$

In general, for a graph moment of order $p$, denoted as $M_p(A)$, we have an expression:

$$M_p(A) = \prod_{q=0}^{p} c^{i_q k_q} A_{k_q j_{q+1}}. \tag{14}$$

which could have as many as $O(pN^2)$ or at least $O(pN)$ nonzero coefficients. Assuming all these nonzero coefficients are $O(1)$, we get $C_f \sim O(pN^q)$ with $1 \le q \le 2$. Thus, in order for the first term in (11) to be small, we need $n > O(C_f^2) \sim O(p^2 N^{2q})$ neurons in the best case, or $n > O(p^2 N^4)$ in the worst case. Additionally, to make the second error term in (11) small, we would need $S > O(nd) \sim O\left(p^2 N^{2q+2}\right)$ samples. $\qquad\square$

For many real world graphs, we have relatively few samples $(S)$ and a large number of nodes $(N)$, using a fully-connected network for learning network moments is nearly impossible. However, note that graph moments are invariant under node permutations. Similar to how Convolutional Neural Networks (CNNs) exploit translation invariance to drastically reduce the number of parameters needed to learn spatial features, graph convolutional networks (GCNs) exploit node permutation invariance, constraining the weights to be the same for all nodes. Additionally, the weights can also not treat neighbors of nodes differently, as neighbors can be permuted too.

The restriction of being permutation invariant also reduces the representation power of a GCN, forcing it to take a very simple form. Namely, the weights of a GCN $w^a$ are simply multiplied into all entries of $A_{ij}$. This architecture is node permutation invariant, but it also uses node attributes to couple the weights to neighborhoods of nodes. Denote $h_i^a$ as the attribute $a$ of node $i$. The output of a GCN follows the formula below:

$$F(A, h)_i^\mu = \sigma \left( \sum_j A_{ij} h_j^a W_a^\mu + b^\mu \right) \tag{15}$$

where $\mu$ denotes the output dimension and $b$ is the bias term. In principle, $A_{ij}$ can be replaced by any general function $f(A)_{ij}$, defined by the propagation rule.

Following the reasoning above, learning nonlinear functions for $F(A)$ requires a lot of data and parameters. It is, therefore, much easier to combine different propagation rules, aka modules related to the generation processes of the graph, such as diffusion operators $D^{-1}A$ and $D^{-1/2}AD^{-1/2}$ and use them instead of only $A$. We also add a node-wise dense layer (which act similar to a GCN, not mixing different nodes) after each of these operators to mix the outputs of these operators.

## B   Experiment details

we generate $5,000$ graphs with the same number of nodes and varying number of links, half of which are from the Barabasi-Albert (BA) model and the other half from the Erdos-Renyi (ER) model. In the BA model, new nodes attach to $m$ existing nodes with likelihood $p_i$ proportional to the degree of the existing node $i$.

$$p_i = \frac{d_i}{\sum_i d_i}$$

The $2,500$ BA graphs are evenly split with $m = 1, N/8, N/4, 3N/8, N/2$. To avoid bias from order of appearance of nodes caused by preferential attachment, we shuffle the node labels. ER graphs are random undirected graphs with a probability $p$ for every link. We choose four values for $p$ uniformly between $1/N$ and $N/2$. All graphs have similar number of links.

For a configuration model [26], the links are generated as follows: Take a degree sequence, i. e. assign a degree $d_i$ to each node. The degrees of the nodes are represented as half-links or stubs. The sum of stubs must be even in order to be able to construct a graph ($\Sigma d_i = 2m$ ). The degree sequence is drawn from the adjacency matrix of the BA graph. Choose two nodes uniformly at random. Connect them with an edge using up one of each node's stubs. Choose another pair from the remaining $2m - 2$ stubs and connect them. Continue until running out of stubs. The result is a graph with the pre-defined degree sequence. We rewire the edges of BA graphs to obtain "fake" BA graphs. The resulting graphs share exactly the same degree distribution, and even mimic the real BA in higher graph moments.

## C   Learning graph moments without residual connections

Our first attempt to combine different GCN modules is to stack them on top of each other in a feed-forward way mimicking multi-layer GCNs. However, our theoretical analysis shows the limited representation power of this design. In particular, no matter how many layers or how non-linear the activation function gets, multiple GCN layers stacked in a feed-forward way cannot perform well in learning network moments whose order is not precisely the number of layers. We observe in our experiments that this is indeed the case.

As shown in Fig 7 shows the test loss over number of epochs for learning first-order (top), second-order (middle) and third-order (bottom) graph moments. We vary the number of layers from 1 to 4

Figure 7: Expressiveness of GCN module *without* Residual Connections: learning first (top), second (middle) and third (bottom) order graph moments with multiple GCN layers and different activation functions. GCN without residual connections fails to learn well when the target graph moments order is greater than the number of layers. With ReLU, sometimes more layers performs even worse. Also, without residuals higher number of layers doesn't always perform as good as when the number of layer matches the order of the moment exactly.

and test with different activation functions including linear, ReLU, sigmoid and tanh. GCN without residual connections fails to learn well when the target graph moments order is greater than the number of layers. With ReLU, sometimes more layers performs even worse. Also, without residuals higher number of layers doesn't always perform as good as when the number of layer matches the order of the moment exactly.

## D    A Note on Graph Attention Networks

The Graph Attention Network (GAT) [31] modifies a message-passing neural network such as GCN by modifying the weights on the edges of the graph. This changes how much each neighbor $j$ of a node $i$ plays a role in the output of node $i$. Assume the input of the GCN layer where the attention layer is added has $F$ features per node, and the output has $F'$ features. Take the $F \times F'$ shared weight matrix $W$ of GCN. Graph attention uses a function $a : \mathbb{R}^{F'} \times \mathbb{R}^{F'} : \mathbb{R}$ to look at similarities of the linear outputs for different nodes

$$e_{ij} = a(Wh_i, Wh_j) \tag{16}$$

However, $e_{ij}$ is only computed for neighbors, meaning that what we really calculate is

$$e_{ij} = a(Wh_i, Wh_j) \circ \hat{A}_{ij} \tag{17}$$

where $\circ$ denotes element-wise multiplication and $\hat{A}$ is the unweighted adjacency matrix of the graph. The function $a$ is implemented as a neural network with a $2F' \times 1$ weight matrix and softmax activation. Now, consider an unweighted graph so that $\hat{A} = A$. The attention mechanism is a function over neighbors only. It takes the output features and passes a linear combination through an activation function $\sigma$ (usually Leaky ReLU)

$$e_{ij} = \sigma \left( \alpha_1 \cdot W \cdot h_i + \alpha_2 \cdot W \cdot h_j \right) \tag{18}$$

where $\alpha_n$ are $F' \times 1$ weight matrices for the attention layer. This output is then also passed through softmax over the neighbors of node $i$. Compared with normal GCN, the attention network is deciding which neighbors should get more weight in the output of node $i$. Our modular approach does not

distinguish among neighbors and is a regular message-passing neural network. We concatenate the output of multiple propagation rules, but each rule is still used in a regular message passing step.