[Reviews · NeurIPS 2019]

Reviewer 1



This paper presents a theoretical analysis of the representational power of graph convolutional network (GCN) models. It is shown that these models are in many cases incapable of learning topological features of the graph such as graph moments, and these shortcomings are used to motivate new GCN architectures combining several propagation rules and incorporating multiple residual connections. The analysis seems to be sound (albeit distant to my area of expertise), and the obtained emprical results seem to support it. I believe that this paper could serve to better improve understanding of GCNs' representational power, and therefore I would vote for (marginal) acceptance. I have a few comments that could be used to improve the work: - In its current form, it is somewhat unclear what "GCN" exactly refers to. It seems that the authors use it interchangeably to refer to the GCN model of Kipf and Welling, and to convolutional layers on graphs more generally. I would recommend a thorough pass through the paper to make it clear what the authors are referring to at different points. - The point that multiple propagation models can be helpful for easing learning has already been empirically pointed out by related work. For example, the graph attention network (GAT) paper (ICLR 2018) finds it critical to use multi-head attention on a graph to achieving strong performance. This effect is probably more pronounced in attention-based models where the propagation model itself is learnable, but it might be worth positioning the proposed architecture in the context of such works as well. - Very similarly, the idea of using residual connections resonates heavily with the jumping knowledge network architecture (Xu et al., ICML 2018), and it would be good to position the work with respect to it as well. - Minor: when referring to the propagation models in sect. 4, I believe the references for Hamilton et al. and Kipf and Welling are flipped with respect to their propagation models. - One thing I found unclear for the graph classification setup is how exactly the graph-level features are computed. The propagation rule explains how to obtain node-level representations, and it's clear how these can be used for node-level classification, but how are these aggregated to a graph-level representations? Is simple mean-pooling used? I invite the authors to clarify. - The experimental analysis is good, thorough and seems to support the theoretical observations - I believe it could be interesting to test the method on more "real-world" graphs as well. Would it make sense to launch the architecture on the standard graph classification datasets (as used in the GIN paper)? --- After rebuttal --- I thank the authors for constructively addressing my comments and providing improvements to the clarity of the setup. I am still unable to fully gauge the contribution of the paper, and the lack of additional benchmarks makes it hard for me to position the proposed operator against prior work (such as GIN). Especially related to the point the rebuttal makes about GATs, I believe that the authors might be overlooking a relevant point: each of the K attention heads are *learnable*, and thus can and often will learn K separate propagation rules. The reason for suggesting real-world benchmarks simply is to show the benefit of choosing K different "hard-coded" propagation models as the authors do here, as opposed to being able to learn them (as GAT/GIN do). I still think the work contains worthy theoretical contribution on an overlooked problem, and I will retain my score.

Reviewer 2



This paper studied an important problem regarding the representation power of GNN when learning graph topology. The paper first introduced graph moments, a key indicator for characterizing the random process of graph generation. Then they showed that a particular variant of GNN with linear activation function and graph filer D^{-1}A has issue in learning degree of a graph. They further pointed the cause of the this limitation due to permutation invariant and then proposed a new modular GCN design. Some theoretical analysis is also provided regarding the graph moments. Some cases study are shown to justify the claims. In general, this paper studied an important problem - understanding the power of GNN regarding its representation since GNN area has been fast growing in recent years. However, there are severe limitations of this paper: 1) It looks to me that authors try to use GNN for graph generation, which graph moments plays a key factor for characterizing this process. Authors pointed out a particular GNN variant with linear activation function and graph filter D^-1A as well as no node attributes has limitations in learning graph moments. However, this particular choices failed does not mean no GNN can do it. Especially, the chosen GNN variants seems like a very uncommon one. 2) Authors tried to connect the failures of a GNN variant in learning graph moments with the property that GNN needs to enforce node permutation invariant. And it seems to me that A_hat = D^-1A caused this issue. However, I have no idea how this conclusion is obtained. Not to mentioned that no GNN paper used D^-1A (but using D^-1/2AD^-1/2), why D^-1 A is associated with node permutation invariant. In fact, the node permutation invariant is typically enforced the node aggregation function (well discussed in GraphSAGE paper). 3) In lines 181-182, authors listed three popular GCN modules and which papers used them (GCN, graphSAGE, and GGNN). However, I did not see these papers used these three different graph operations, instead they only used f3 = D^-1/2AD^-1/2. I am a little supervised how authors get this conclusion. For the proposed GCN module, I did not see why these designs can help learn graph moments. Also, this design is just a simple variant of original GNN and I did not see any novel thing here. These designs have been used in many applications papers. 4) The experimental evaluations are weak. They are many graph generation papers in the literature both from conventional literature and recent GNN papers. But I didi not see any baselines there.

Reviewer 3



GCNs are widely used and well studied in both theoretical and application works. Although this work is not the first one to analyze the representation power of GCNs in learning graph topology, it still brings in some fresh air in the field of graph neural networks. It has potential impact on researchers working on GCNs and related applications, as a guideline of how to design the architecture of GCNs to improve the representation power. The paper is well-written and easy to follow. Experiments demonstrate that the proposed modular design can increase the representation power of GCNs.

Reviewer 4



Summary ======= The authors investigate the capability of graph convolutional networks (GCN) to approximate graph moments. Herein a graph moments is a polynomial functions of the graph's adjacency matrix. In a theoretical section they analyze necessary conditions on GCNs to be capable of learning such moments. The main insight here is that: (1) Exact depth configuration, i.e., a deep enough architecture selection, is necessary to learn a graph moment of certain order (degree of the polynomial). (2) Residual connections allow to learn also moments of lower order than the maximum order which can be achieved with the selected architecture's depth. In the light of this considerations the authors propose a unified residual architecture combining three different message passing paradigms. Experimentally, the authors conduct an ablation study where they evaluate the impact of depth, wideness, and choice of activation function on the performance of regressing different graph moments. The theoretical findings support the results of those experiments. In another experimental part the authors evaluate similar architectural design choices in a classification setup on synthetic data (i.e., random graph generation). Originality =========== The theoretical result are interesting and original. Quality ======= Except for one issue the overall quality of this work is good, could, however, by polished a little more to improve readability and clarity (see below). The main concern is the lack of related work concerning graph moments. Are they introduced by the authors? If not, state where they are originally defined! Do they have application so far? Given that this work is built around graph moments this has to be handled, one way or the other. Clarity ======= Notation: There should be done some improvement concerning the mathematical notation. The main concern is a lack of explicit introduction of notational convenience which makes things unnecessarily hard to understand, e.g., 1) p3 l112 what is M_p? shouldn't this be M_p(A) 2) sometimes you use an index such as in f(A)_i, I think you mean the i-th coordinate function? 3) The moments are matrix valued, right? 4) p3 l92 why two times \hat{A}? 5) (other things of similar nature) Contribution: It seems that the theoretical contribution is the core of this work. However, the proposed unified layer (using 3 message passing paradigms simultaneously) and the introduction of residual connections are very strongly/(over?) stated and even get an experimental section. But I do not understand the motivation to present this here as contribution. For me, it is not clear how the theoretical findings motivate to gather several message passing paradigms in one *module*. Significance ============ The work is potentially significant. Even more if the importance of moments for graphs could be stated more clearly (see Quality --> related work comment).

[Author Response · NeurIPS 2019]

We sincerely thank the reviewers for their insightful and constructive comments. First, we address the common
questions of **R1,R2,R3:** *Definition of GCN:* We define GCN in eq (4), which not only includes Kipf & Welling [16]
$D^{-1/2}AD^{-1/2}$, but also $D^{-1}A$ (e.g. [12] GraphSAGE mean aggregator) among others. Other papers (e.g. JK paper by
Xu et al, 2018, p3) also consider these as variants of GCN. We will make this explicit in the final version. *Contribution:*
the main contribution of this work is to develop deep theoretical understanding of GCNs, which would inform efficient
GCN architectures design. We do not intend to achieve the state-of-the-art graph classification model. We will tone
down the emphasis on the model architecture design, as similar work may already exist. *Real-world Experiment:* To
verify theoretical results, we needed the ground truth model, which is unknown for real data. We are happy to add
real-world experiments in the final version. Next, we address the specific concerns raised by each reviewer below.

**R1** *graph attention network (GAT)* multi head attention in GAT concatenated $K$ independent attention mechanism
with the *same* propagation rule (PR) while ours utilizes modules with *different* PRs. We will consider incorporating
attention mechanism as a future work. *jumping knowledge network* Both JK-network and ours use residuals, but ours
uses residual connections for GCN while JK-network is designed for Message Passing Graph Neural Networks. *graph*
*classification setup* We used mean-pooling to aggregate node-level representations, after which a single number is
passed to a classification layer. We will clarify this and include discussions with GAT and JK in the final version.

**R2** *authors try to use GNN for graph generation* There seem to be some misunderstandings about the paper. We do *not*
learn graph generation. Instead, we use GCN to learn graph moments and to classify graphs. *... $D^{-1}A$ has limitations*
*in learning degree of a graph.* We are *not* claiming the limitation of a particular GCN variant (Fig 2 is only an example).
Our key message is that unlike fully connected neural networks, GCNs are not universal approximators. Therefore,
choosing the right PR (for example, $A$ vs $D^{-1}A$) is crucial in the GCN's ability to learn graph moments. We provide
theoretical analysis and offer a solution that can alleviate this issue. Note that $D^{-1}A$ is *not* our definition of GCN (see
eq (4)), nor is it used in our theoretical analysis. *... why $D^{-1}A$ is associated with node permutation invariant.* This is
a misunderstanding. We are *not* claiming that $D^{-1}A$ is related to node permutation invariance. In fact, any GCN in
the form of eq. (4) $F(A, h) = \sigma(f(A) \cdot h \cdot W + b)$ is permutation invariant, regardless of the function $f(A)$ (see sec
2.1 and 2.3). The purpose of Proposition 1 and Theorem 2 is to argue that GCN can be restrictive due to permutation
invariance, thus having the right PR, activation and number of layers is crucial in its ability to learn graph moments.

*did not see these papers used these three different graph operations* It is not explicit but easy to derive. [12] GraphSAGE
with MEAN aggregator, averages over $h_i + \sum_{j \in \mathcal{N}_i} h_j$, which is equivalent to the operator $\tilde{D}^{-1}\tilde{A}$ where $\tilde{A} = A + I_N$ is
an adjacency with self-loops. Kipf-Welling GCN uses $\tilde{D}^{-1/2}\tilde{A}\tilde{D}^{-1/2}$ ([16] eqs (7),(8)). [18] uses the graph module $A$
([18] eq (2)) *simple variant of original GNN* Our theoretical analysis demonstrates the importance of PR (e.g. using $A$,
GCN cannot learn $\sum_j (D^{-1}A)^p_{ij}$ or vice versa), as well as to have sufficient number of layers with residual connections.
Hence, having a modular design with sufficiently many layers and residual connections would be useful for learning
graph moments. We have no intention to claim the novelty of our proposed GCN, but only to validate our theoretical
findings. In fact, we pointed out the similarities our model with GIN in lines 173-176. *graph generation baselines* As
we are *not* generating graphs in this work, our method is unrelated to graph generation baselines.

**R3** *better not to claim...expressiveness of GCNs as the first contribution.* While it may be known in the literature,
we are not aware of any rigorous theoretical analysis for the exact same setting. We will add more references and
tone down the claim in the final version. *bridge the theoretical analysis with the proposed design.* Our theoretical
analysis shows the limitations of having a single PR in learning graph moments. It also points out the importance of
having sufficient number of layers and residual connections. Our modular design combines these results and arrive at
an architecture with multiple PR modules, layers and residual connections. We will improve the writing and include
more descriptions in the final version. *compare with some existing GCN designs* Our GCN already includes several
existing GCN designs, which we refer as different PRs. We are happy to include other GCN designs for comparison as
well. *Could not find the code...* We apologize for the confusion. We will release the code in the updated version.

**R4** *lack of related work concerning graph moments.* Graph moments, or "Graph Power" is a concept from graph
theory ( see the Lin and Skiena (1995) ) and has been used extensively in network science. Graph moments encodes
topological information of a graph and is closely related to graph coloring and Hamiltonicity. We will include references
from graph theory and network science. Note that we are not aware of any other work that learns graph moments using
GCNs. *Notation: lack of explicit introduction of notational convenience* Thank you, will fix. *Line 112* yes, for brevity
$M_p$ is $M_p(A)$. *an index such as in $f(A)_i$.* Yes, $f(A)_i$ is the $i$th component. *The moments are matrix valued?* No, our
definition eq. (1) is vector valued, (summed over one node index). *residual connections are very strongly stated...* We
agree with the reviewer regarding the positioning of our contribution. We will tone down the emphasis on modular
design. The restrictions found in our theory show a need for including multiple propagation rules, hence the module.

[Meta-Review · NeurIPS 2019]

Unfortunately reviewer_2 did not engage in the discussion even though it was needed, my recommendation is thus mainly based on the 3 other reviews. This paper investigates the expressive power of Graph neural networks by studying to which extent they can compute graph moments. This is an interesting and original approach and the theoretical findings are sound and relevant to the community. In preparing the camera-ready version of the paper, the authors should take the following points in consideration. The reviewers mentioned that the exposition of some of the contributions are somehow overstated and should be tuned down. Experimental evaluations on real data and comparisons with existing GCNs (even though the purpose of the paper is not to beat SOTA, this would make an informative addition) would improve the paper. Lastly, the paper could be polished with a thorough proof reading to improve its clarity.